# Methods of Detecting Medication Administration Point-of-Care Errors in Acute Adult Inpatient Settings: A Scoping Review Protocol

**DOI:** 10.3390/mps5020032

**Published:** 2022-04-14

**Authors:** Julie-Anne Martyn, Angela Ratsch, Kaye Cumming, Jennifer Dredge

**Affiliations:** 1School of Nursing, Midwifery, and Paramedicine, University of the Sunshine Coast, Hervey Bay 4655, Australia; Jennifer.Dredge@nt.gov.au; 2Research Services, Wide Bay Hospital, and Health Service, Hervey Bay 4655, Australia; angela.ratsch@health.qld.gov.au; 3Rural Clinical School, The University of Queensland, Bundaberg 4670, Australia; k.cumming@uq.edu.au

**Keywords:** medication administration errors, detection methods, acute, adult, hospitalized

## Abstract

Medication administration is recognized as a risk-prone activity where errors and near misses have multiple opportunities to occur along the route from manufacturing, through transportation, storage, prescription, dispensing, point-of-care administration, and post-administration documentation. While substantial research, education, and tools have been invested in the detection of medication errors on either side of point-of-care administration, less attention has been placed on this finite phase, leaving a gap in the error detection process. This protocol proposes to undertake a scoping review of the literature related to the detection of medication errors at the point-of-care to understand the potential size, nature, and extent of available literature. The aim is to identify research evidence to guide clinical practice and future research at the medication and patient point-of-care intersection. The search strategy will review literature from PubMed, CINAHL, Cochrane Collaboration, Embase, Scopus, PsychInfo, Web of Science, TRIP, TROVE, JBI Systematic Reviews, Health Collection (Informit), Health Source Nursing Academic, Prospero, Google Scholar, and graylit.org dated 1 January 2000–31 December 2021. Two independent reviewers will screen the literature for relevancy to the review objective, and critically appraise the citations for quality, validity, and reliability using the Joanna Briggs scoping review methodology and System for Unified Management, Assessment and Review of Information (SUMARI) tool. The data will be systematically synthesized to identify and compare the medication error administration detection method findings. A descriptive narrative discussion will accompany the findings.

## 1. Introduction

Medication errors can cause physical and psychological harm to patients, psychological distress for healthcare personnel, and fiscal burdens to health services [1,2]. Worldwide, substantial education, resources, tools, devices, and systems have been invested in preventing medication errors [3]. An equivalent approach has been taken to measuring and reporting errors to demonstrate the effectiveness of the error preventive strategies and to identify gaps in the processes [4,5]. The measurement and reporting of medication errors is an expected safety and quality standard for accredited Australian healthcare providers [6].

However, the measurement and reporting of a medication error is reliant upon detection. Globally, a range of detection processes have been established which seek to recognize errors in the manufacturing, transportation, storage, prescription, and dispensing phases [7,8]. Pre- and post-administration medication detection is primarily centered upon the use of auto-technology functions in electronic medical records, electronic prescribing, and/or chart audits, the latter often occurring some days to weeks (or longer) after the medication administration [9,10]. The chart audit process is informative for health services in terms of system deficits and educational gaps, however, it is not a ‘live’ tool able to detect and prevent errors at the point-of-care administration.

At the point-of-care, detection of a medication administration error (MAE) is the last barrier to the avoidance of a medication error. Safety guidelines, including the universal five rights mantra, ‘right patient, right medication, right dose, right route and right time’, have evolved over the past 50 years as a standard aspect of point-of-care medication administration [11]. Abiding by these guidelines is promoted as a control mechanism, however, the evidence that these guidelines are an effective detection method at the point-of-care medication administration phase is absent [12].

Direct observation is one method said to produce reliable MAE point-of-care evidence [13], nevertheless, direct observation for detecting medication errors is not without its complexity due to its multifaceted construct. For example, a systematic review that was limited to direct observational studies found disparities in the data collection methods and study settings which hampered the validity and reliability of observation as an error detection method [5]. Internationally, there appears to be no standard operating procedure for medication administration or any evidenced-based and comprehensively consistent method to detect MAEs at the point of administration. Consequently, the measurement and reporting of such errors are essentially unreliable.

This scoping review aims to identify, describe, and synthesize literature related to the detection of MAEs at the point-of-care in acute adult inpatient settings. The review will identify key definitions and concepts, procedures utilized in clinical practice, types and sources of evidence to inform practice and policy, and gaps in the research. The factors affecting medication administration will be noted: for example, time of day, place of administration, person/s involved, type of administration, and patient characteristics. The findings will provide greater insight into the factors influencing the procedure at the phase where MAEs intersect with the patient.

### Definitions

Globally, the rate of reported MAEs is high [14,15,16]. However, MAE is an ill-defined construct resulting in an extensive collection of research, education tools, and protocols to avoid such errors [17,18,19], with inconsistent approaches to the detection of MAEs including varying data definitions, data inclusion, and collection methods, and an equally prolific body of resources and literature surrounding the measurement and reporting of MAEs. In addition, studies often fail to consider or report the method of MAE detection. Accordingly, the source of evidence is not clear. Without robust, standardized MAE detection methods at the point-of-administration, reported MAE rates are unreliable.

Conceptually, the medication point-of-care is nuanced, the perspective of which is defined by the healthcare provider’s role in the medication process [7]. A prescriber’s point-of-care is usually the medication order and could exclude the patient; likewise, the pharmacist’s point-of-care starts with the preparation and dispensing of medications and may never include interaction with the patient [7]. The medication administration point-of-care occurs with a direct interaction between the healthcare provider and the patient. This exact point of care is of interest because there is a lack of interrogation of this MAE intersection in the literature.

In this review, this point of the process will be isolated from the pre-and post-administration phases and will only consider MAEs and their detection at the exact point-of-care when the prescribed medication is administered to the patient. In this review, the following definition of medication error will be used to determine an MAE at the administration point-of-care, “A medication error is a failure in the treatment process that leads to, or has the potential to lead to, harm to the patient” [20] (p. 599). This broad definition is often used in MAE literature (in its original and adapted form) to define and classify MAEs [7].

## 2. Materials and Methods

### 2.1. Objectives and Methodology

The aim of the study is to identify what research methods are used to detect errors at the medication administration point-of-care of patients in acute adult inpatient settings. This scoping review will examine the literature describing the methods used in acute, adult inpatient settings to detect MAEs at the point-of-administration and explain how they detect MAE at that intersection. To preface this review, a search of PROSPERO, Research Registry, MEDLINE, the Cochrane Database of Systematic Reviews, and the Joanna Briggs Institute Database of Systematic Reviews and Implementation Reports was conducted on 17 March 2022. No completed, commenced, or proposed scoping or systematic reviews on this topic were identified. The proposed scoping review will be conducted following the Joanna Briggs Institute methodology for scoping reviews [21]. The data extraction process will follow the JBI SUMARI protocol [22], collecting the publication details, the study design and details, and the specific concepts of interest for this scoping review as outlined in Appendix B and Appendix C.

This scoping review will consider any literature that provides information related to the detection methods for MAEs at the point of administration. It is expected that the literature included in this scoping review will provide information related to:MAE rates and prevalence;Factors that contribute to MAEs;Descriptions of medication administration types and practices;Intervention strategies to avoid and manage MAEs.

### 2.2. Concepts and Context

The concept of interest is the method of detecting medication administration errors at the point of care. The participants of interest are any person, including the patient, involved in administering medications. This review will be restricted to literature related to MAE in acute care adult hospital settings regardless of geographic location. The restriction to this context is intended to limit the variance in the medication administration and management processes identified in other settings such as community, pediatrics, aged care, and mental health. The processes to administer medications in those settings involve context-specific protocols such as double-checking and identification by a photo that is not typical in hospital acute care settings [23,24,25].

### 2.3. Literature Sources: Inclusion and Exclusion Criteria

The review will consider full-text original research, including quantitative, qualitative, and mixed methods studies, doctoral theses, and gray literature that include all the scoping review concepts listed in Appendix A. Text and opinion papers will also be considered for inclusion if they meet the inclusion criteria. Literature published in English and between 1 January 2000–31 December 2021 will be included. The date range restriction relates to the release of the publication called *To Err is Human: Building a Safer Healthcare System* [26]—a publication that triggered international patient safety hypervigilance in healthcare [27]. While not specific to this study, the aforementioned foundational work initiated a paradigm shift in healthcare safety—of which medication error detection, management, and avoidance is a significant component.

Literature focusing on medication errors other than those detected at the medication administration point-of-care will be excluded. Specifically, it is the methods to detect medication administration point-of-care errors that is the phenomenon of interest. Therefore, the literature that describes medication administration point-of-care errors without explaining the findings from a detection method will also be excluded. Conference proceedings and abstracts will be excluded, as will commercial literature or profiteering organization promotional materials. Literature reporting on prescription, transcription, or dispensing errors will not be included unless the error is detected at the point-of-medication administration to the patient.

### 2.4. Participants

This review will consider literature that reports on medication errors and includes adults in acute hospital settings and any person involved in administering medications to hospitalized patients. The healthcare provider could consist of doctors, pharmacists, nurses, midwives, allied health professionals, and their assistants. The review will also consider any reported patient or family involvement in administering medications in the inpatient setting.

### 2.5. Search Strategy

The search strategy will locate published and unpublished studies, reviews, text, and opinion papers. An initial limited search of PubMed was undertaken to identify literature on the topic. The title words, abstract, keywords, and index terms were used to develop a full search strategy (Appendix A). The search strategy will be adapted for each included information source. The reference lists of articles selected for full-text review will be screened for additional papers that meet the inclusion criteria (Appendix B and Appendix C). 

The information sources will be retrieved from literature databases, health industry websites, public documents, and theses. The databases to be searched include PubMed, CINAHL, Cochrane Collaboration, Embase, Scopus, PsychInfo, Web of Science, TRIP, TROVE, JBI Systematic Reviews, Health Collection (Informit), Health Source Nursing Academic, Prospero, Google Scholar, and graylit.org. The search of Google Scholar will be conducted using the keywords: acute, adult, hospitalized, medication errors, point-of-care medication administration, and detection methods to a page depth of 10. Sources of unpublished studies and gray literature will be searched using greylit.org using the same keywords. Additional unpublished studies and grey literature will be sourced on the following health websites: World Health Organization, Safe Medication Practice Unit, Australian Commission on Safety and Quality in Healthcare, and NHS Improvement.

Following the search, the identified citations will be collated and uploaded into EndNote 20.2.1 [28], and duplicates removed. Two independent reviewers will then screen the citation titles and abstracts and assess them against the review inclusion criteria. Potentially relevant papers will be retrieved in full and their citation details imported into the Joanna Briggs Institute’s System for the Unified Management, Assessment, and Review of Information (JBI SUMARI) [22]. 

### 2.6. Data Extraction and Presentation

The full text of selected citations will be assessed in detail against the inclusion criteria (Appendix B and Appendix C) by two independent reviewers. Reasons for excluding full-text papers that do not meet the inclusion criteria will be recorded and reported in the final scoping review. Any disagreements between the reviewers at each stage of the selection process will be resolved through discussion or with a third reviewer. The search results will be reported in full in the final scoping review and presented in a Preferred Reporting Items for Systematic Reviews and Meta-analyses extension for Scoping review (PRISMA-ScR) flow diagram [21]. The quality, validity, and reliability of each citation will be assessed using the relevant JBI critical appraisal tool that is integrated into the JBI SUMARI software [22]. 

The results of this scoping review will identify the detection methods for MAEs at the medication administration point-of-care in acute adult inpatient settings. The healthcare personnel involved will be identified, and how the errors were detected and reported will be described. The factors affecting medication administration such as the types and practices will be described. The extracted data will be presented in tabular form and categorized in a manner that aligns with the concepts and objectives of the review question. A narrative summary including a metanalysis of the data will accompany the results and describe how the results relate to the review’s objective and question.

## 3. Expected Results and Implications

This scoping review will map literature that describes medication errors at the medication administration point-of-care where the medication and patient intersect. The review will describe the literature details, including research designs, participant characteristics, study context, error types, and error detection methods. The researchers will systematically synthesize the literature to identify and compare the MAE detection method findings. A descriptive narrative discussion including a metanalysis of the data will accompany the findings. It is expected that this scoping review will identify several methods for detecting medication administration point-of-care errors. However, there may be limited homogeneity amongst the methods and hence, the research findings are of limited use. The researchers anticipate results to inform future practice and research at the medication administration point-of-care interaction in adult patient acute care settings. These results could guide future developments of more rigorous, systematic, and real-time methods to detect medication administration point-of-care errors, thus reducing their occurrence. 

## Data Availability

The data presented in this study are available on request from the corresponding author. The data are not publicly available due to ethical constraints of research.

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
