# Peer review of "Methods of Detecting Medication Administration Point-of-Care Errors in Acute Adult Inpatient Settings: A Scoping Review Protocol"

_mps, 2022, doi:10.3390/mps5020032_

Round 1

Reviewer 1 Report

Thank you for the opportunity to read this Manuscript. I consider this manuscript as innovative and important, especially to those with closely related topics. The purpose of the paper is well defined, the methods are appropriate, the data is sound and overall the manuscript adheres to the relevant standards for reporting. However, I have a few concerns and suggestions for major and minor revisions:

  • In methods, it is not cleare how study selection and data extraction will be conduct.
  • What about data coding?
  • How you want to assess the quality of study included?
  • Please, provide a brief summary of attended results and improve the “implication for practice” section with more details.

Reviewer 2 Report

Overall, the paper is well written, clear and detailed.  It provides all the required details with regards to the methodology for conducting the scoping review; the databases being searched are comprehensive, search terms are appropriate, and eligibility of studies is clearly drafted.  My only recommendation would be to expand on the data being abstracted.  I do not see particulars related to medication errors that may be of importance in the analysis.  For example, type of formulation - oral vs. injectable, timing of medication administration (shift change versus within shift), number of patients under care, etc - these may be factors that could affect medication errors at point of administration.  

Reviewer 3 Report

The study aims to undertake a systematic review of methods of detecting point-of-care MAE for acute adult inpatients. The topic is important for healthcare industries and worth conducting a systematic review. Overall speaking, the protocol for this review is sufficient and feasible. Two minor suggestions for the authors’ reference.

1.Since this study will collect MAE rates and prevalence, will quantitative meta-analysis be conducted? It is not clear now.

2.Since the target is the adult patient, it will be more clear to explicitly specify the age range in the inclusion/exclusion criteria.
